# Review of the Oscillation of Research Regulations for Bioethics in the Republic of Korea: Comparison with Japan

**DOI:** 10.3390/biotech12020047

**Published:** 2023-06-15

**Authors:** Seung-Hyo Hyeon, Juyoung An, Hwa-Shin Ryoo, Min-Kyu Lee

**Affiliations:** 1Department of Public Administration, Graduate School, Chungbuk National University, 1 Chungdae-ro, Seowon-gu, Cheongju-si 28644, Chungcheongbuk-do, Republic of Korea; po@chungbuk.ac.kr; 2Faculty of Policy Science, Ryukoku University, 67 Tsukamoto-cho, Fukakusa Fushimi-ku, Kyoto 612-8577, Japan; juyoung@policy.ryukoku.ac.jp; 3Law School, Chungbuk National University, 1 Chungdae-ro, Seowon-gu, Cheongju-si 28644, Chungcheongbuk-do, Republic of Korea; lawdeo@chungbuk.ac.kr; 4Department of Public Administration, College of Social Sciences, Chungbuk National University, Chungdae-ro, Seowon-gu, Cheongju-si 28644, Chungcheongbuk-do, Republic of Korea

**Keywords:** Bioethics Act, Woo-seok Hwang, Shinya Yamanaka, stem cell

## Abstract

The Bioethics Act in the Republic of Korea has undergone great fluctuations akin to the pendulum of a clock. Since Professor Hwang’s research ethics issue, domestic embryonic stem cell research has lost its vitality. This study argues that the Republic of Korea needs a reference point that does not waiver. This study examined the characteristics of life science- and ethics-related systems in the Republic of Korea and Japan. It also examined the pendulum-like policy changes in the Republic of Korea. It then compared the strengths and weaknesses between the Republic of Korea and Japan. Finally, we proposed a system improvement strategy for the development of bioethics research in Asian countries. In particular, this study argues that the advantages of Japan’s slow but stable system should be introduced.

## 1. Introduction

The Japanese government has been active in resolving ethical, legal, and social issues regarding human embryonic stem cell and embryonic research, in what has been classified as a research-friendly policy [1]. However, in Japan, there is also a view that government regulations could hinder various areas of human stem cell research. In particular, as an Asian country, Japan is criticized for its inability to make quick decisions, even though it has a centralized government. The development of stem cell research policies and regulations in Japan has involved lengthy periods of discussion, preparation, and review, taking five to ten years for each case. These regulatory delays have presented challenges to Japanese researchers, hindering their progress and competitiveness. Japan has had limited involvement in human embryonic stem cell (hESC) research, compared to other countries, due to regulatory delays and a lack of guidelines for the international distribution of hESC lines. Regulatory developments have also hindered Japan’s participation in somatic cell nuclear transfer (SCNT) and germline differentiation studies, limiting researchers to animal studies [2]. 

Although Japan is a country without religious or political confrontation regarding bioethics, it has been criticized for slow decision making compared to other countries, such as Singapore, due to bureaucracy [3]. Japan’s bioethics system is slow and stuffy. Japan has achieved a successful case of research with the development of groundbreaking technologies such as iPS (induced Pluripotent Stem) cells [4]. The use of iPS cell technology raises ethical and legal concerns regarding the informed consent of tissue donors, but it is considered to raise fewer concerns compared to the use of embryonic stem (ES) cells [5].

In contrast, the Republic of Korea’s bioethics legislation has experienced great fluctuations akin to the swinging of a pendulum. The Bioethics Act passed by the National Assembly (the congress) in 2003 reflected the opinions of the government and members of the National Assembly, who insisted that life science and technology should be developed. In July 2002, after announcing that the Korean branch of Clonade was conducting research on human cloning, the Ministry of Health and Welfare of the Republic of Korea made a pre-announcement of the bioethics bill. The bill prohibited the creation of embryos for purposes other than pregnancy and allowed research on embryos older than five years after in vitro fertilization. At a public hearing held on October 9th of the same year, the scientific community generally agreed, but civic groups objected [6,7,8]. It was evaluated that the Korean government listened to the stance of civic groups in the early stage, but it sided with the scientific community in the final stage [6].

In 2004, Dr. Woo-seok Hwang succeeded in obtaining ES cells through SCNT technology in human eggs, and in 2005 he reported that he had created “customized cloned human embryonic stem cells”. However, the journal Nature pointed out problems in relation to the research, such as the provision of eggs by female researchers and the review by the Hanyang University Clinical Ethics Committee in May 2004. The Korean Society for Bioethics sent an open inquiry requesting the sources of 242 eggs for Dr. Woo-seok Hwang’s research [9].

After Woo-seok Hwang’s research ethics issue was highlighted, the opinion that the social atmosphere should strengthen bioethics has gained strength. Since bioethics was emphasized in the revision of the Bioethics Act in 2018, ES cell research in the Republic of Korea lost vitality. In the revision of the Bioethics Act in 2020, requirements for acceptance were eased by reflecting the opinions of the scientific community again [9].

We compare the regulations related to life science research in the Republic of Korea and Japan. In particular, an issue arose in the Republic of Korea because its regulations were insufficient to ensure ethics and safety. We argue that the Republic of Korea needs to take advantage of Japan’s slow but stable system, which we will introduce later. We also review the process that has hindered the development and discuss the desired direction for the development of life science research.

## 2. Review of Research-Related Systems for Life Sciences in Japan

### 2.1. Background on the Establishment of Laws Related to Human Embryonic Stem Cells in Japan

In 2000, the Japanese National Diet (the congress) enacted the “Act on the Regulation of Human Cloning Technology and Other Technologies” and prohibited human cloning. Regarding human embryonic research, in September 2001 the “Guidelines for the Establishment and Use of Human Embryonic Stem Cells” was announced, allowing human embryonic research under certain restrictions. In June 2004, the Bioethics Committee under the Ministry of Education, Culture, Sports, Science, and Technology allowed human embryonic research only for basic science research, but not for clinical application. In July 2006, the Bioethics Committee published the “Guidelines for Clinical Research Using Human Stem Cells” and banned clinical research [10].

### 2.2. Research Trends and Achievements in Japan

In Japan, there was research on introduced pluripotent stem (iPS) cells, which are relatively free from ethical issues related to the use of eggs and embryos [10]. In 2006, Shinya Yamanaka Shinya’s team succeeded in generating iPS cells from mouse embryonic or adult fibroblasts [4]. In 2007, they succeeded in generating iPS cells from adult humans [11]. Yamanaka was awarded the Nobel Prize in Physiology or Medicine for this achievement in 2012. The Japanese government actively supports research and development using iPS cells. In June 2013, clinical trials using iPS cells were approved in Japan for the first time in the world [10]. In 2013, researchers in Japan succeeded in generating iPS cells from adults using a combination of plasmids encoding OCT3/4, SOX2, KLF4, L-MYC, LIN28, and shRNA for TP53, which are easily accessible. It is expected that making iPS cells from less invasive tissues would facilitate disease treatment [12].

### 2.3. Japan’s Slow but Stable System for Life Science Research

In May 2014, Japan renamed the Council for Science and Technology (CST), which was established during the reorganization of the government organization, to the Council for Science, Technology and Innovation (CSTI), strengthening the regulatory function over science and technology policy [10]. The CSTI is chaired by the Prime Minister under the control of the Cabinet Office. In principle, the CSTI meets at least once a month. The characteristics of the CSTI are “strategic and timely”, “comprehensive”, and “voluntary”. “Strategic and timely” means that a comprehensive strategy related to science and technology should be established to respond to national and social challenges in a timely manner. “Comprehensive” emphasizes the relationship between society and humans, such as ethical issues including humanities and social sciences. “Voluntary” means not only responding to the advice of the Prime Minister and others but also expressing one’s own opinion. The prime minister, as well as related ministers, researchers, and lawmakers, actively participate in the CSTI meeting, and the detailed minutes of each meeting are made public. Materials referenced by members are made public on the Home Office website [13].

The Bioethics Professional Investigation Society (BPIS) was established under the CSTI. The BPIS reviews the guidelines for the establishment and use of human ES cells. The BPIS held its first meeting in 2001 and the 136th meeting on 27 Feb 2023. Fifteen meetings were held over a period of about one year and nine months to discuss how to amend guidelines allowing clinical research on human ES cells. At the 75th meeting held in September 2013, trends in gamete generation research were reviewed. At two meetings held in October and November of the same year, opinions on the latest research trends were presented by researchers. For about one year and six months from December 2013 to June 2015, whether to allow research on human ES cells was discussed (Table 1) [14,15].

## 3. Review of Research-Related Systems for the Life Sciences in the Republic of Korea

### 3.1. Review of Life Science Technology and Bioethics-Related Systems in the Republic of Korea

The Ministry of Health and Welfare and the Ministry of Science and Technology each prepared a bill to enact the Bioethics Act. The Ministry of Health and Welfare announced the draft in December 2000 to collect the opinions of civic groups that were emphasizing bioethics [16] (pp. 45–47). In May 2001, the Ministry of Science and Technology produced the basic framework of the Basic Act on Bioethics, which was different from the Ministry of Health and Welfare’s plan in that it prohibited the cloning of human embryos [6] (pp. 56–57). The announcement of this basic framework caused an organized backlash from the scientific community [16] (p. 56).

In 2001, the Citizen Science Center of the People’s Solidarity for Participatory Democracy formed a network with religious, women’s, environmental, and animal rights groups that judged that the bioethics bills would be difficult to pass due to opposition from scientists. On 19 July 2001, they officially launched the “joint campaign group for the prompt enactment of the Basic Act on Bioethics” [16] (pp. 56–60). In December 2002, the American company Clonade claimed to have created a cloned baby. At this time, members of the National Assembly submitted bills to ban human cloning [6] (pp. 61–62).

Eventually, the government’s final draft was passed in the plenary session of the National Assembly on 29 December 2003. The government finally sided with the scientific community and put more emphasis on fostering biotechnology [6] (p. 45), [17] (p. 167).

In the process of legislating the Bioethics Act, opinions in favor of the ethics community were discussed first. However, it ended with the scientific community and the ethics community confronting each other. Due to this confrontation, various actors in the policy network contributed strongly by quickly adjusting their interests. However, by focusing only on solving the problem quickly, a debate within the scientific community about how to perform specific technology in accordance with bioethics was ignored. The fact that the process of thinking and discussing was omitted remains a problem.

### 3.2. Discussions on the Bioethics Act in the Republic of Korea

#### 3.2.1. Discussion of the Moral Status of the Human Embryo

The author of a law thesis divided a human embryo’s status into personalism and impersonalism. They critiqued impersonalism from a personalist perspective, arguing that an embryo should be protected like a human since it can never transition from non-human to human [18,19].

It was also argued that a human embryo should be considered equal in moral status to an adult, even before being implanted in the womb. Thus, using cloned embryos for experimentation created through in vitro fertilization and SCNT was criticized as an act that undermines the dignity of human life. This argument advocated for the cessation of such experiments [20].

Human life extension has become a reality through medical advancements, but issues with organism cloning have emerged. It has been argued that treatment with adult stem cells poses minimal ethical concerns, while using ES cells raises ethical dilemmas due to harm inflicted on the embryo. Furthermore, considering the continuity and potential of life, there are no justifiable reasons to prioritize human life over embryonic life [21].

#### 3.2.2. Discussion from a Feminist Perspective

The feminist point of view argued that the existing concept of life did not deviate from the patriarchal and male-centered point of view and that women’s voices and experiences were ignored in biotechnology. [22].

Human eggs can be obtained only through a women’s donation, as artificial production is not yet possible. However, the process of superovulation using ovulation injection can lead to physical discomfort and even life-threatening symptoms for women. Therefore, the argument highlights the importance of handling egg usage in biotechnology research with caution and respect for women’s well-being [23].

The media’s coverage of Woo-seok Hwang incident was criticized from a feminist perspective. The feminist media focused on human rights-based bioethics and criticized the nationalist approach that treated women’s bodies as tools for life science. In contrast, the mainstream media prioritized a utilitarian discourse highlighting national interests and creating a divide between “advanced science” and “outdated ethics”, and marginalizing women’s perspectives [24].

#### 3.2.3. Discussions from a Legal Perspective

The law governing life sciences and biotechnology was criticized for its broad and abstract provisions. The focus was on the use of oocytes, which are cells involved in oogenesis. Concerns were raised about the potential exploitation of oocyte donors by researchers. It was argued that women donating oocytes for medical or reproductive purposes should be afforded extra protections. Additionally, during the revision of the Bioethics Act, there were calls for provisions regulating stem cell research and cross-species transplantation [25].

The institutionalization of bioethics in the Republic of Korea was criticized as inadequate. It was argued that participation in bioethics discussions should extend beyond bioethicists, scientists, and lawyers to include scholars from other fields. Furthermore, it was emphasized that a rationalist model should be pursued to establish public ethics [26].

### 3.3. Reinforcing Life Science Research Regulations after the Woo-seok Hwang Incident

#### 3.3.1. Research Misconduct by Woo-seok Hwang’s Team

The “Woo-seok Hwang Incident” occurred after the enactment of the Bioethics Act [7]. In 2004, Dr. Woo-seok Hwang reported that his team had succeeded in obtaining ES cells through SCNT technology in human eggs. In 2005, he reported that he had created “customized cloned human embryonic stem cells” [8,27]. In a May 2004 special article in *Nature* revealed that eggs were provided by a doctoral student in Dr. Woo-seok Hwang’s team and another female researcher. Researchers are inevitably vulnerable to pressure from research directors. Thus, it was believed that egg donation by the researchers was inappropriate [28]. After raising these issues, on 22 November 2005, MBC PD Notebook aired with the theme of “Suspicion of Woo-seok Hwang’s myth” [27].

It was pointed out that Dr. Woo-Seok Hwang’s team violated the guidelines prohibiting the creation of human embryos for research purposes while receiving eggs from female researchers. The Institutional Bioethics Board (IRB) of Hanyang University took responsibility for reviewing and approving the research protocol. Questions were also raised as to whether this was conducted properly. This pointed out that the IRB system and organization, which had left bioethics to the conscience of the researchers was also responsible [29].

#### 3.3.2. Various Opinions after Woo-seok Hwang Incident

As a result of the apology for the incident, Deputy Prime Minister Myung Oh, who served as the Minister of Science and Technology, resigned and appointed Deputy Prime Minister Woo-shik Kim, while Kim acknowledged the need for human embryo cloning and stem cell research, saying, “First of all, the problem is that the focus is on performance, and then I think that the problem of research ethics, integrity, and insufficient verification systems have worked in combination”. It became an issue in local elections in 2006, as well as the presidential election in 2007. The position of the Grand National Party candidate Myung-bak Lee, Uri Party, Democratic Party, and the People First Party were in the position to allow an exception for the purpose of treating rare and incurable diseases, while the position of the Democratic Labor Party candidate Young-gil Kwon was to ban it completely [9].

In the investigation into the Woo-seok Hwang incident, it was found that somatic ES cells could not be produced even after using about 2000 eggs [27]. Regarding this, in July 2006, a “Discussion on the Reevaluation of Somatic Cell Cloning Embryo Research” was held at Ewha Womans University, where stakeholders such as domestic stem cell researchers and bioethicists gathered and discussed. Discussions were focused on the issue of stem cells and the possibility of research on somatic cell cloning of embryos. At this forum, the bioethics community took the position that concerns about bioethics had increased after the incident. Participants expressed concerns that female eggs might be indiscriminately donated for somatic cell cloning. In particular, in Woo-seok Hwang’s case, most egg donors were family members of patients with incurable diseases and 15 to 20% of them developed hyperovulation syndrome. Ra-geum Huh, a professor at Ewha Womans University, argued that this practice should be corrected [30]. Protestants, Catholics, women’s groups, and civic groups opposed somatic ES cell research from the perspective of damaging human dignity [9].

In contrast, the position of embryo cloning researchers was that there was no need to reconsider the decision to allow somatic cell-cloning embryo research from two to three years ago. Hyeong-min Jeong, a professor at CHA University argued that only the Republic of Korea was regressing at a time when foreign scientists has started research on somatic ES cells. Professor Dong-wook Kim of Yonsei University also took the position that “now it is more important to discuss the scope of permission rather than whether or not to permit research”. However, although some scientists agreed with the position of emphasizing bioethics, they were aware of the concern that human eggs should not be used indiscriminately. Professor Hyeong-min Jeong said that it was right to apply it to humans after conducting sufficient animal research. Professor Dong-wook Kim and Professor Yong-man Han of the Republic of Korea Institute of Science and Technology (KAIST) also said that bioethics education and publicity were necessary for researchers. It was the position that consciousness needed to be strengthened [30]. In a situation where the possibility of technological success was slim, there was a coexisting position that it was difficult to allow embryo cloning research without any safety measures (Table 2) [9].

#### 3.3.3. Revision of the Bioethics Act in 2008

In the midst of such conflicting opinions, an amendment to the Bioethics and Safety Act was passed by the National Assembly on 16 May 2008. This bill was a combination of the main contents of the Grand National Party lawmaker Jae-Wan Park and the government amendment bill. It was aimed at protecting the health of egg donors by conducting health examinations and limiting the frequency of egg collection [31].

The revision of the Bioethics Act in 2008 after the Woo-seok Hwang incident appeared to be strengthening the act. Regarding research on somatic cell cloning of embryos, which was an issue, the existing limited permission was maintained, while the range of eggs that could be used for research was limited, and the health protection of egg donors was further strengthened [8].

The revision of the Bioethics Act in 2008 ensured the safety of egg donors, expanded the scope of the prohibition on interspecies SCNT research, and established the Institutional Bioethics Review Committee to be set up in institutions that perform research on life science technology. Its purpose was self-regulation by implementing support for the regulation [9].

### 3.4. Deregulation to Promote Life Science Research

#### 3.4.1. Opinions of the Scientific Community after the Revision of the Bioethics Act in 2008

Since the revision in 2008, the scientific community has consistently raised concerns that the scope of research allowed on gene therapy is too narrow [9]. The Ministry of Science and ICT (MSIT) jointly held the 9th Bio Economic Forum at the National Assembly with Yong-hyeon Shin, a member of the People’s Party, and discussed the direction for revising the Bioethics Act. In that forum, experts identified three major problems with the Bioethics Act: positive regulation, a comprehensive prohibition that blocked both basic research and clinical trial research, and centralized regulation. In addition, there was an opinion that the procedure for obtaining research permission was very difficult [32].

Life science researchers have pointed out that the current Bioethics Act is blocking innovative research and development (R&D). Researchers agreed that “regulation by disease” of gene therapy (clinical) research should be abolished and that it was not reasonable to limit the content of embryonic research by law. Researchers agreed on the need to allow basic research, eliminate overlapping regulations, consider changes in technology and environment, and expand autonomy at research sites. In addition, opinions suggested that “differentiated regulation” was needed according to the research topics and degree of violation of bioethics, and that the National Bioethics Committee should cooperate with IRBs of private institutions to recognize management based on autonomy [33].

There seemed to be no major disagreement about amending the provisions of the law at the time that limited the diseases subject to research on somatic cell gene therapy. Instead, some called for a system to monitor and manage risks that may appear after gene therapy is implemented. So-ra Park, a Professor of Medicine at Inha University (physiology), said, “It is necessary for the scientific community to monitor themselves, educate and train themselves, and establish guidelines such as open research. It is also necessary to operate an Ethics Committee, and it is also necessary to open the discussions of the Ethics Committee to the outside world [34]”.

Seung-joon Yoo, Director of the Republic of Korea Center for Bio-Economic Research, said, “For the clinical application of medical technology, it is necessary to allow the generation of embryos for research and research at the level of major countries (of research and development). In this case, it seems necessary to take strong penalty measures such as punitive damages [34]”.

At the time, research on gene-related treatments and embryos was fundamentally blocked except for 22 specified rare and incurable diseases. Scientists saw these restrictions as excessive and demanded that they be lifted at the United Kingdom, United States and Japan levels. In particular, they insisted on changing from ‘positive regulation’, which specifies the names of diseases allowed and restricts all other cases, to ‘negative regulation’, which explicitly limits only those to be restricted. Even legal scholars and regulatory researchers are generally in agreement on this point [35].

There were also criticisms about the reality of having to go through multiple approvals and reviews as the review agencies and procedures overlapped. Professor Dong-ryul Lee criticized, “If you go through the Institutional Review Board (IRB) and the National Bioethics Committee (NBC), the deliberation continues for more than a year at most, and studies that have been abandoned because of this [35]”.

#### 3.4.2. Opinions of the Bioethics Community after the Revision of the Bioethics Act in 2008

In particular, the issue of embryonic research could be amplified into a controversy over bioethics when it coincides with the religious world’s view of life. Professor Jae-woo Jeong of Catholic University (Dean of the Graduate School of Life Sciences) said, “Creating embryos for research means creating weak human beings in need of protection and nurturing to be used as a research tool, and this cannot be tolerated. It is not a matter to be decided by majority vote [34]”.

Instead of allowing research, it was pointed out that strict management is needed to secure human rights and research ethics in the process of obtaining and using embryos and reproductive cells. This was because there were continuous criticisms that the IRB was performing only perfunctorily [32].

Some suggested that sensitive ethical issues should be dealt with through public debate involving experts and citizens. Professor Hyeon-cheol Kim of Ewha Womans University Law School said, “The question of how far embryos will be allowed for research is inevitably a major issue of conflict”, adding, “We need a public debate with citizen participation [24]”. He also argued, “The bioethics law should be left as the basic law, and the research itself should be treated separately as an individual law [35]”.

#### 3.4.3. Revision of the Bioethics Act in 2020

Conditions for permitting research on gene therapy were partially reflected in the 2020 revision, and conditions for permitting gene therapy research were alleviated. However, revisions, such as obligatory review by the institutional committee for research plans, and so on, were made. This law was proposed to ease the requirements for permitting research on gene therapy so that more diverse research on gene therapy could be conducted in the Republic of Korea, not to supplement the risks that may occur due to the relaxation of the permitting standards with the institutional committee review system (Table 3) [9].

## 4. Discussion

### 4.1. Comparison of Life Science Research Regulatory Policies in the Republic of Korea and Japan

Due to Japan’s unique bureaucratic nature, participation of various actors in the policy-making process is not guaranteed. Decision making is not fast either. However, it is possible to make specific decisions with expertise through the participation of experts. Life science researchers, who can be called regulated subjects, can make predictions. It has the advantage of being able to provide possible and actionable guidelines.

In the Republic of Korea, from the beginning of the establishment of the Bioethics Act, has had conflicting characteristics with confrontation between the scientific community and the ethical community. Activities of government departments to secure regulatory authority have occurred. This characteristic has made policy makers interested in whether or not to allow research in relation to life science and technology regulation. However, progress has been hampered due to the lack of specific discussions on how to allow such research.

The excessive permissibility of research led to the Woo-seok Hwang incident. In 2008, due to strong demands from civil society, the regulation of the Bioethics Act was strengthened, resulting in a decline in research. After nearly ten years of excessive regulation, criticism from the scientific community intensified again and regulations were eased in 2020. In this way, the Republic of Korea’s life science research regulations have fluctuated akin to a pendulum (Table 4) [15].

### 4.2. Desired Direction of Life Sciences Regulatory Policies in Asia

The advantages of the network related to life science and bioethics in the Republic of Korea include the guaranteed participation of various actors and quick decision making. Therefore, for the development of life science- and bioethics-related systems and organizations in the Republic of Korea, it is necessary to accept Japan’s strengths without losing the Republic of Korea’s strengths. The Republic of Korea’s Bioethics Act was enacted in a way that allowed too much research. As a result, life science researchers have deviated and the level of regulation of the Bioethics Act has increased, hindering research development.

By accepting the advantages of Japan’s slow but stable system, researchers will not act in a way that undermines bioethics. Asian countries, in particular, need to introduce organizations such as Japan’s BPIS, where government officials and scientists go through a deliberation process to improve life science research regulations.

In-depth discussions centered on the government and experts, which are Japan’s strengths, could enable concrete decision making with expertise for the development of systems and organizations related to life science and bioethics in Asian countries. Sufficient issues should be discussed and data should be provided to whole communities. Through this, it is possible for life science researchers to recognize predictable and practicable action guidelines in research and to equip religious groups, women’s groups, civic groups, bioethics groups, and others with expertise for activities and sufficient monitoring.

This process will ensure bioethics and safety in Asian countries and ultimately contribute to the development of life science research.

### 4.3. Futher Discussion and Future Work

The Republic of Korea’s rapid decision making and Japan’s slow but stable decision making system research regulations were compared. It was also argued that the Republic of Korea should accept the merits of Japan’s decision-making system. However, it was a limitation that this paper only compared these two Asian countries.

It was difficult to precisely compare and analyze life science research achievements and their economic effects in the Republic of Korea and Japan. In the future, these two countries will need to actively conduct such research.

However, in the Republic of Korea, the prevailing opinion is that the Republic of Korea is far ahead of Japan in science and technology. The number of Nobel laureates in the field of science symbolizes the level of basic science and original technology. Japan has 24 Nobel Prize winners, but the Republic of Korea has none. It is a lamentation that there are winners from the neighboring country, but not from the Republic of Korea, which is also an Asian country [36,37].

Since this paper only compared the Republic of Korea and Japan, comparing life science policies in other Asian countries will be an important future research task. In particular, China is a country that should be examined.

The CRISPR-Cas9 genome editing system, derived from bacterial adaptive immune strategies, is a powerful tool for precise modification of the target genome in living cells, allowing control over functional genes with high accuracy [38]. However, due to its powerful nature, this tool might raise ethical concerns, such as the loss of human dignity. Furthermore, it has the potential to lead to catastrophic events, such as the spread of unintended mutations in the human gene pool.

For example, Chinese researcher He Jiankui, known for his claim of creating genetically edited babies, was found guilty of conducting illegal medical practices and sentenced to three years in prison. He and his collaborators were found to have forged ethical review documents and misled doctors into implanting gene-edited embryos [39]. Dr. He has been found guilty of forging approval documents and deceiving couples in a trial held in Shenzhen. He claimed to have prevented human immunodeficiency virus (HIV) infections in newborns through gene editing but was found to have misled both the subjects and medical authorities. Dr. He’s controversial work resulted in the birth of twin girls and an undisclosed third genetically edited baby [40].

It will be an important task to quantitatively identify the relationship between life science and technology policy regulation and socioeconomic effects. After that, we will be able to discuss the socio-economic effects of expanding our system to other Asian countries.

Some conditions must precede the introduction of such a system in Asian countries. First, the authority of scientists should be secured so that an atmosphere in which the public and policy makers can accept scientists can be created. The second is to overcome the superiority of politics and administration over the field of science and technology.

## Figures and Tables

**Table 1 biotech-12-00047-t001:** List of Bioethics Professional Investigation Society (BPIS) conferences related to human ES cell research.

No.	Date	Title
89–78	3 June 2015 (Heisei 27) - 20 December 2013 (Heisei 25)	▪ Regarding research on the production of human embryos by germ cells generated from human ES cells, etc. ▪ Regarding the status of the reexamination of relevant guidelines for human ES cells ▪ Regarding the review status of the revision of guidelines for human ES cells ▪ Other matters
77	27 November 2013 (Heisei 25)	▪ Listening to trends in germ cell generation research, such as ES cells: Atsuo Ogura (Director, Bioresource Center, Institute of Physical and Chemical Research) and one other person
76	18 October 2013 (Heisei 25)	▪ Listening to trends in germ cell generation research, such as ES cells: Takehiko Ogawa (Professor, Department of Molecular Biomedical Sciences, Department of Medicine, Yokohama City University)
75	20 September 2013 (Heisei 25)	▪ Regarding the trend in germ cell generation research, such as ES cells

Source: [14,15].

**Table 2 biotech-12-00047-t002:** Conflicts of positions in the 2008 revision of the Bioethics Act.

Division	Expansion of Regulations	Reduction of Regulations
Participants	Protestants/Catholics, civic groups, women’s groups, Democratic Labor Party	Life scientists, Woo-seok Hwang support group, Buddhists
Faith	Bioethics, prohibition of embryo research	Improve national competitiveness, permission to study embryos for research and therapeutic purposes
Policy preference	- Agree with the revision of the Bioethics Act - Residual embryo research and the production and research of somatic cell cloning of embryos are prohibited - Expansion of adult stem cell transplantation - Prohibition of xenogeneic nuclear transfer - Genetic testing is prohibited in principle	- Opposition to the revision of the Bioethics Act - Elimination of restrictions on the type of eggs used for research - Preparation of grounds for allowing oocyte donation for treatment and research purposes and compensation for actual expenses - Allow cross-species experiments - Withdrawal of free stem cell provision

Source: [9].

**Table 3 biotech-12-00047-t003:** Comparison of the Bioethics Act and its revisions.

Act	Main Contents	Regulation Level
2004 Enactment	Establishment of the Presidential Advisory Council on Science & Technology (PACST). Establishment of the Institutional Bioethics Review Board (IRB) at institutes with embryo research, gene banks, and gene therapy institutes. The implantation, maintenance, or birth of cloned embryos in the womb for the purpose of human cloning was prohibited. The production of embryos for purposes other than conception was prohibited. Somatic cell nuclear transfer for purposes other than research for the treatment of rare or incurable diseases, etc., was prohibited.	The permissible range of research was wide and the system to prevent deviance was insufficient.
2008 Revision	Mandatory health checkup for egg donors. The frequency of oocyte retrieval was limited. Somatic cell nuclear transfer between humans and animals was prohibited. The use of stem cell lines was permitted only for purposes such as research for diagnosis, prevention, and treatment of diseases.	The permissible range of research was narrow enough to discourage research.
2020 Revision	Relaxation of acceptance conditions for research on gene therapy.	The permissible range of research was wide and a system was in place to prevent deviance.

Source: [9].

**Table 4 biotech-12-00047-t004:** Comparison of life science research regulatory policies in the Republic of Korea and Japan.

Division	Republic of Korea	Japan
Policy actors	The range of actors is wide and diverse with the participation of government, science, and ethics.	The government and experts are at the center, and civil society participation is weak.
Policy change	At first, the level of regulation was low, but after the Woo-seok Hwang incident, it fluctuated.	Regulatory change is slow.
Advantages	▪ The participation of a large number of actors has been guaranteed, and rapid decision making has been made amid conflicting issues.	▪ Able to make professional and specific decisions. ▪ It can provide predictable and actionable action guidelines for life science researchers.
Disadvantages	▪ Rationality and expertise in the process and content leading to the policy are somewhat lacking.	▪ Due to its bureaucratic nature, the participation of many is not guaranteed, and decision making is not fast.
Implications	▪ It is desirable to maintain the strengths of our system while embracing the strengths of the Japanese system. ▪ It is necessary to enable detailed decision making with expertise through in-depth discussions centered on the government and experts. ▪ Asian countries, in particular, need to introduce organizations such as Japan’s BPIS, where government officials and scientists go through a deliberation process to improve life science research regulations.	

Source: [15].

## Data Availability

This research did not generate or analyze any data.

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
