# Peer review of "Review of the Oscillation of Research Regulations for Bioethics in the Republic of Korea: Comparison with Japan"

_biotech, 2023, doi:10.3390/biotech12020047_

Round 1
Reviewer 1 Report
The authors could address potential economic or social impacts of introducing Japan's slow but stable system to other Asian countries, which would provide a more holistic view of the proposed system improvement strategy.
The authors could identify potential challenges or barriers to implementing their proposed system improvement strategy and provide recommendations for overcoming them. This would help ensure that their proposed strategy is feasible and practical in real-world settings.

See Report
Author Response
a) As you pointed out, it would be valuable for the authors to consider the potential economic and social impacts of introducing Japan's slow but stable system to other Asian countries. However, it is unfortunate that there is a lack of quantitative research comparing the research performance and economic effects of Korea and Japan based on their regulatory systems. It is necessary for such research to be conducted more actively in the future. Nevertheless, as mentioned in the revised manuscript, it is generally recognized that Korea still lags behind Japan in terms of scientific and technological achievements (Section 4.3,. [36-37] pp. 10-11). Taking this into account, it would be desirable for Asian countries to consider adopting Japan's system.
b) As stated in the revised manuscript's conclusion, certain conditions must precede the introduction of such a system in Asian countries. Firstly, the authority of scientists needs to be established, creating an environment where the public and policymakers can accept their expertise. Secondly, the dominance of politics and administration over the field of science and technology must be overcome(Section 4.3, p.11)
c) Thank you for your feedback and suggestions. We greatly appreciate them, as they have contributed to the further development of our paper. Once again, thank you very much.

Reviewer 2 Report
Thank you for the hard work. Very interesting topic. Would the author also discuss the next step for this research/topic?
Not sure if this topic is interested enough to attract more readers.
Author Response
a) The research compared Korea's rapid decision-making with Japan's slow but stable decision-making system in research regulations, arguing that Korea should adopt the merits of Japan's system. However, the paper had a limitation in only comparing these two Asian countries, and more precise comparisons and analysis of life science research achievements and economic effects are needed. Comparing life science policies in other Asian countries, particularly China, will be an important future research task (Section 4.3, p.10-11).
b) Quantitatively identifying the relationship between life science and technology policy regulation and socioeconomic effects is an important task to discuss the expansion of such systems to other Asian countries (Section 4.3, p.10-11).
c) Thank you for your feedback and suggestions. We greatly appreciate them, as they have contributed to the further development of our paper. Once again, thank you very much.

Reviewer 3 Report
Very interesting work, the only drawback is, in my opinion, that the comparison includes only two systems (Japan and Korea).
Minor harsh English use, but satisfactory.
Author Response
a) While this study did not provide a detailed analysis of Asian countries other than Korea and Japan, it mentioned the potential for expansion to other Asian countries, including China, in future research (Section 4.3, p.10-11).
b) Future research should include a comparative analysis of life science policies in other Asian countries, with particular emphasis on China. Chinese researcher He Jiankui, known for his claim of creating genetically edited babies, was found guilty of conducting illegal medical practices, forging ethical review documents, and deceiving medical authorities. These developments highlight the need to consider ethical implications and potential risks associated with the application of genome editing technologies in human beings (Section 4.3, p.10-11).
c) Furthermore, in future research, the effects of adopting Japan's slow but stable system in other Asian countries, as well as the key challenges that need to be addressed for its implementation, were discussed (Section 4.3, p.10-11).
d) Thank you for your feedback and suggestions. We greatly appreciate them, as they have contributed to the further development of our paper. Once again, thank you very much.

Reviewer 4 Report
- The argument is interesting, but it should be improved. I recommend a broader, more detailed analysis and a more argumentative discussion.
- The Introduction Section has to be better argued and more detailed.
- The Bibliography must be extended. References are quite limited for a Review, therefore it is necessary to find more papers that raise debates on this topic.
- Section 3.3.1. Opinions of the scientific community after the revision of the Bioethics Act in 2008, should be developed and better scientifically justified. I recommend the extension of this section, with more bibliographic sources including the opinions of several scientific researchers in the field of Life Science on this topic, information that would probably clarify more the points of view of the participating groups involved in the debate on this theme, groups mentioned by the authors (groups of women, civic, religious, political groups, bioethics groups, etc.)
Author Response
a) In the introduction, a more detailed discussion was conducted, emphasizing the legislative focus on scientific and technological advancement over bioethics in Korea. Furthermore, the argument against the notion that Japan's slow but stable system hinders scientific and technological advancement was refuted using the example of induced pluripotent stem cells (iPS), particularly highlighting the lower ethical concerns associated with iPS compared to embryonic stem cells (ES). ([5], pp. 1-2).
b) In order to expand the scope of the discussion, additional references were cited in the study. Particularly, to further elaborate on the ethical perspectives of Korea, such as the ethical community and women's perspectives, a new Section 3.2 was added, building upon the original manuscript's Section 3.3.1. This section summarized the views in relation to bioethics (pp. 4-5). Additionally, in Sections 3.4.1 and 3.4.2, the opinions of the scientific and ethical communities after 2008 were included to argue the hindered development of life science and technology in Korea following the Hwang incident (pp. 7-8).
c) Thank you for your feedback and suggestions. We greatly appreciate them, as they have contributed to the further development of our paper. Once again, thank you very much.

Round 2
Reviewer 2 Report
NA
NA